# Remodelling of the Mitochondrial Bioenergetic Pathways in Human Cultured Fibroblasts with Carbohydrates

**DOI:** 10.3390/biology12071002

**Published:** 2023-07-14

**Authors:** Margherita Protasoni, Jan-Willem Taanman

**Affiliations:** Department of Clinical and Movement Neurosciences, UCL Queen Square Institute of Neurology, University College London, Royal Free Campus (M12), Rowland Hill Street, London NW3 2PF, UK; margherita.protasoni@irbbarcelona.org

**Keywords:** fructose, galactose, glucose, mitochondria, mitochondrial DNA, mitochondrial respiration, oxidative phosphorylation

## Abstract

**Simple Summary:**

Many neurological diseases are caused by defects in the powerplants of cells, called mitochondria. To study these diseases in the laboratory, researchers often grow skin cells (fibroblasts) derived from patients in a dish. Glucose is normally used to feed the skin cells. However, when skin cells are fed with glucose, they rely less on mitochondria to generate energy but prefer glycolysis as alternative energy-generating pathway. As a result, defects in mitochondria are not showing because the skin cells are not using the powerplants. Therefore, these diseases cannot be studied with skin cells grown in a dish with glucose. We investigated if skin cells can be fed with two other sugars, galactose and fructose. We show that although the skin cells grow more slowly, they are using their mitochondria to generate energy when fed with galactose or fructose. Thus, these neurological disorders can be studied in a dish if skin cells from patients are fed with galactose or fructose. As skin cells grow much slower in a dish when fed with galactose than when fed with fructose, we recommend feeding skin cells with fructose to study these diseases the laboratory.

**Abstract:**

Mitochondrial oxidative phosphorylation defects underlie many neurological and neuromuscular diseases. Patients’ primary dermal fibroblasts are one of the most commonly used in vitro models to study mitochondrial pathologies. However, fibroblasts tend to rely more on glycolysis than oxidative phosphorylation for their energy when cultivated in standard high-glucose medium, rendering it difficult to expose mitochondrial dysfunctions. This study aimed to systematically investigate to which extent the use of galactose- or fructose-based medium switches the fibroblasts’ energy metabolism to a more oxidative state. Highly proliferative cells depend more on glycolysis than less proliferative cells. Therefore, we investigated two primary dermal fibroblast cultures from healthy subjects: a highly proliferative neonatal culture and a slower-growing adult culture. Cells were cultured with 25 mM glucose, galactose or fructose, and 4 mM glutamine as carbon sources. Compared to glucose, both galactose and fructose reduce the cellular proliferation rate, but the galactose-induced drop in proliferation is much more profound than the one observed in cells cultivated in fructose. Both galactose and fructose result in a modest increase in mitochondrial content, including mitochondrial DNA, and a disproportionate increase in protein levels, assembly, and activity of the oxidative phosphorylation enzyme complexes. Galactose- and fructose-based media induce a switch of the prevalent biochemical pathway in cultured fibroblasts, enhancing aerobic metabolism when compared to glucose-based medium. While both galactose and fructose stimulate oxidative phosphorylation to a comparable degree, galactose decreases the cellular proliferation rate more than fructose, suggesting that a fructose-based medium is a better choice when studying partial oxidative phosphorylation defects in patients’ fibroblasts.

## 1. Introduction

The two major interconnected metabolic pathways in the cell that produce energy in the form of ATP are glycolysis in the cytosol and oxidative phosphorylation (OXPHOS) in the mitochondria. During glycolysis, glucose is broken down into pyruvate through several steps that yield a small amount of ATP. Under anaerobic conditions, pyruvate is converted into lactate and secreted by the cell. Under aerobic conditions in the body, pyruvate is normally imported into the mitochondria where it is decarboxylated into acetyl-CoA, which is fed into the Krebs cycle. NADH and FADH_2_, produced in the Krebs cycle, are oxidised by enzyme complexes of the OXPHOS system. The free energy of the redox reactions is used to maintain a proton electrochemical gradient across the inner mitochondrial membrane, which is exploited to drive ATP synthesis. This OXPHOS process is the main source of ATP in the body [1]. However, when cells are cultivated in medium with standard, supraphysiological concentrations of glucose (25 mM), mitochondrial ATP production is suppressed even in the presence of abundant oxygen. This glucose-induced inhibition of OXPHOS is known as the Crabtree effect [2].

Mitochondrial disorders affect one in ~2000 individuals [3]. Dermal fibroblast cultures are often the only tissue material readily available from patients with a mitochondrial dysfunction; however, cultivation in customary high-glucose medium often masks the defect in culture due to the Crabtree effect. To overcome this problem, fibroblasts have been cultivated in medium in which glucose was replaced with galactose, which quells glycolysis and stimulates OXPHOS. Although cells with severe OXPHOS defects will not survive when galactose is substituted for glucose [4,5,6,7], milder OXPHOS defects are expressed in a more pronounced manner [8,9,10,11]. In addition, medium containing galactose instead of glucose has been used to test for mitochondrial toxicity of drugs [12,13,14,15].

Glucose as well as galactose and fructose are abundant monosaccharides metabolised by the cell. Galactose is metabolised from lactose (milk sugar), a disaccharide of glucose and galactose, whereas fructose is metabolised from sucrose (table sugar), a disaccharide of glucose and fructose. Galactose enters the glycolytic pathway after its stepwise conversion into glucose-1-phosphate through the Leloir pathway [16]. In the body, fructose is trapped in the liver as fructose-1-phosphate and converted into fat [17]. In cultured cells, however, fructose enters the glycolytic pathway via phosphorylation by hexokinase to fructose-6-phosphate [18]. Unlike glucose, which is predominantly metabolised to pyruvate after entering the glycolytic pathway, galactose and fructose are largely metabolised through the oxidative arm of the pentose phosphate pathway, generating NADPH and ribose-5-phosphate, a precursor of nucleic acid synthesis [18,19]. In a medium containing galactose or fructose in place of glucose, energy is provided by glutamine [18,20], which is typically present at a concentration of 4 mM in standard medium. Glutamine is first hydrolysed to glutamate by glutaminase, followed by conversion to the Krebs cycle intermediate α-ketoglutarate by glutamate dehydrogenase [21]. Thus, glutamine provides anaplerotic replenishment of the Krebs cycle, thereby generating NADH and FADH_2_ as substrates for OXPHOS.

Although others have studied metabolic adaptations of cells cultivated in medium in which galactose replaced glucose [15,18,19,22,23,24,25,26,27], studies concerning fructose have been limited [18,19,28]. Moreover, nearly all these studies were conducted with cancer cells or differentiating muscle cells, which might respond differently to carbohydrate changes in the medium than skin fibroblasts. Here, we report a systematic study of the adaptations of cultured human skin fibroblasts to the substitution of glucose in the medium with galactose or fructose. As highly proliferative cells depend more on glycolysis than less proliferative cells [29], we compared proliferation rates of a fast-growing and a slower growing fibroblast culture derived from a neonatal and an adult donor, respectively. Thirteen of the ~90 subunits of the OXPHOS enzyme complexes are encoded on the mitochondrial DNA (mtDNA) [30]. Therefore, we measured mtDNA copy numbers to investigate if mitochondrial gene dosage plays a role in metabolic remodelling. In addition, we assessed transcript and protein levels of glycolytic enzymes, markers of mitochondrial content, mitochondrial transcription factors, OXPHOS enzyme subunits, and assembled OXPHOS enzyme complexes, along with enzyme activities and mitochondrial respiration. Collectively, our results indicate a small increase in mitochondrial content and a large, disproportional increase of OXPHOS enzymes when glucose in the medium is replaced with galactose or fructose.

## 2. Materials and Methods

### 2.1. Cell Culture Conditions

A primary human skin fibroblast culture was established from a healthy 27 year old female according to standard conditions [31]. The donor provided prior informed written consent. Ethical approval was obtained from the Royal Free Hospital and Medical School Ethics Committee (REC07/H0720/161) in compliance with national legislation and the Declaration of Helsinki. A neonatal human control skin fibroblasts culture (106-0n) was purchased from the Health Protection Agency (0609717).

Cells were cultivated at 37 °C in a humidified atmosphere of 5% CO_2_ in air, in Dulbecco’s modified Eagle medium (DMEM) containing 25 mM d-(+)-glucose and 4 mM l-glutamine (Gibco, 11965092), or DMEM without glucose but with 4 mM l-glutamine (Gibco, 11966025), supplemented with either 25 mM d-(+)-galactose or 25 mM d-(-)-fructose. All three media were supplemented with 10% dialysed foetal bovine serum, 1 mM sodium pyruvate, 0.2 mM uridine, 50 units/mL of penicillin, and 50 μg/mL of streptomycin. To ensure that the cells were fully adapted to the carbohydrate source, cultures were maintained in medium with a particular carbohydrate for ≥3 doubling times prior to analyses. Medium was refreshed every 3 days and on the day before harvesting. Confluency was maintained at <90%. To harvest the cultures, cells were dislodged by trypsinisation, washed with medium, followed by two washes with phosphate-buffered saline (PBS). Cell pellets obtained by centrifugation were stored at −80 °C and used for experiments within 1 week.

### 2.2. Growth Rate

Trypsinised fibroblast suspensions in the three types of media were counted on C-Chip haemocytometer slides (NanoEnTek, Seoul, Republic of Korea), followed by seeding of each culture in triplicate in five 12-well plates at a density of 2000 cells/well in 500 μL of medium. Cultures growing in glucose-based medium were incubated for 1–5 days, whereas cultures growing in galactose- or fructose-based medium were incubated for 2–10 days. For cultures in glycose-based medium, one plate was aspirated each day and stored at −80 °C, while for cultures in galactose- or fructose-based medium, one plate was aspirated and stored every other day. When a plate was stored, media in the wells in the remaining plates was replenished by replacement of 200 μL of medium. When all plates were collected, the amount of double-stranded DNA in each well was determined with the CyQUANT Cell Proliferation Assay (ThermoFisher Scientific, C70026, Waltham, MA, USA) and compared with that of a serial dilution of 0–50,000 cells to calculate the number of cells per well [32]. Doubling time (DT) of the cultures was calculated with the equation: DT = (t − t_0_)log2/logN − logN_0_(1)
where t and t_0_ are the times at which the plates were stored, and N and N_0_ are the cell numbers at those times.

### 2.3. Copy Number of mtDNA

DNA was isolated from cell pellets with the Puregene Core Kit A (Qiagen, 1042601, Hilden, Germany) and quantified with the Qubit dsDNA HS Assay Kit (ThermoFisher Scientific; Q32854) on a Qubit Fluorometer (ThermoFisher Scientific). The mtDNA copy number per diploid nuclear genome (i.e., per cell) was determined by multiplex droplet digital PCR (ddPCR), using BioRad’s QX200 ddPCR system as recommended by the manufacturer. Primer and TaqMan probe sequences (Eurofins, Luxembourg) for amplification and detection of regions of the mtDNA D-loop and nuclear *B2M* gene [33] are shown in Appendix A. For measurements, 1.6 ng of total genomic DNA was mixed with 0.9 μM of each primer, 0.25 μM of each TaqMan probe, and 1× ddPCR Supermix for Probes (BioRad Laboratories, 186–3010) in a final volume of 192 μL, followed by the generation of droplets with a BioRad QX200 Droplet Generator. Next, 20 μL samples were activated at 95 °C for 10 min, amplified during 40 cycles of denaturation (94 °C, 30 s) and annealing/extension (61 °C, 60 s), followed by a final denaturation (98 °C, 10 min), using a BioRad C1000 Touch thermal cycler. Finally, droplets were read with a BioRad QX200 Droplet Reader and analysed with Biorad QuantaSoft software. Samples were examined in octuplicate.

### 2.4. Reverse Transcriptase Real-Time Quantitative PCR

RNA was extracted from cell pellets with the RNeasy Mini Kit (Qiagen, 74104), quantified with a NanoDrop spectrophotometer (ThermoFisher Scientific, Horsham, UK) and cDNA was reverse transcribed from 1 μg of RNA with the QuantiTect Reverse Transcription Kit (Qiagen, 205311). Real-time quantitative PCR (RT-qPCR) was performed with PowerUp SYBR Green PCR Master Mix (ThermoFisher Scientific, A25780) as described [34]. Some primers were designed using Primer-BLAST (NCBI) and purchased from Eurofins, whereas other primers were directly ordered from Qiagen (Appendix A). Transcript levels of cells cultured with galactose or fructose were quantified relative to those of cells cultured with glucose [34]. Normalisation was carried out using β-actin transcript levels. Experiments were performed 5–7 times.

### 2.5. Western Blot Analysis

Cell pellets for sodium dodecyl sulphate (SDS)-denaturing gel electrophoresis were extracted with 1% Triton X-100 and samples (10 μg of protein/lane) were analysed on Western blots as described [35], except that samples were resolved on MiniProtean TGX 4–20% gels (BioRad Laboratories, Watford, UK, 4568096) alongside a Precision Plus Protein Standard (BioRad Laboratories, 1610374). Each blot was re-probed with anti-β-actin antibodies to verify loading.

Cell pellets for blue-native gel electrophoresis were lysed in 1.5% n-dodecyl-β-d-maltoside, 1 M 6-aminocaproic acid, 50 mM bistris (pH7.0), 1 mM phenylmethanesulphonyl fluoride, 1 μg/mL leupeptin, and 1 μg/mL pepstatin A on ice for 15 min, followed by centrifugation at 16,000× *g*, 4 °C for 15 min. Extracts were diluted to an equal protein concentration and ⅙ volume of 1 M 6-aminocaproic acid, 5% Serva blue G (Serva Electrophoresis, 3505003, Heidelberg, Germany) was added prior to loading on 3–12% native polyacrylamide gels with a 3% stacking gel (10 μg of protein/lane) as devised by Schägger [36], using the Mini-PROTEAN System (BioRad Laboratories). After electrophoresis, gels were electro-blotted onto Immobilon-PSQ PVDF membranes (Millipore, 1SEQ00010, Burlington, MA, USA) [37], rinsed three times with methanol to remove residual dye, blocked with 10% skimmed milk powder in PBS, and developed with antibodies as described [35]. To confirm even loading, SDS to a final concentration of 2% was added to the samples, followed by gel electrophoresis, blotting, and development with anti-β-actin antibodies.

The amount of protein in the extracts was determined with the BCA Protein Assay Kit (ThermoFisher Scientific, 23225). Antibodies used for detection are listed in Appendix A. Signals were captured with a BioRad Chemidoc MP Imaging System, quantified with Bio-Rad Image Lab 5.1 software and expressed relative to the β-actin signal. Experiments were performed at least in triplicate with independent samples. Uncropped images are shown in Appendix A.

### 2.6. Enzyme Activity Measurements

The Lactate Dehydrogenase Activity Assay Kit (Sigma-Aldrich, Poole, UK, MAK066) was used to measure lactate dehydrogenase activity of LDH assay buffer (kit) cell extracts, in quadruplicate, as recommended by the supplier. Cytochrome-*c* oxidase activity of the n-dodecyl-β-d-maltoside extracts prepared for blue-native gel electrophoresis was determined in quadruplicate and citrate synthase activity of the Triton X-100 extracts prepared for denaturing gel electrophoresis was determined in triplicate as described [35].

### 2.7. Extracellular Flux Analysis

Mitostress tests were performed on a Seahorse XFp platform (Agilent Technologies, Milton Keyes, UK) as described [32], except that 15,000 fibroblasts were seeded per well of an XFp cell culture miniplate and cells were assayed in Seahorse XF base medium (Agilent Technologies, 102353-100) pH 7.4 (NaOH), supplemented with 10 mM glucose or 10 mM galactose or 10 mM fructose, 1 mM sodium pyruvate, and 2 mM glutamine. After the measurements, results were normalised according to the cell number per well determined with the CyQUANT Cell Proliferation Assay kit [32]. Technical triplicates of the stress tests were repeated 5–6 times.

### 2.8. Statistical Analyses

Graphs and statistical analyses were executed with GraphPad Prism software. Data are presented as mean ± standard error of the mean (SEM). To examine statistical significance, two-way ANOVA with Tukey’s multiple comparison test or with two-tailed ratio paired Student’s *t*-test for pairwise comparisons was used.

## 3. Results

### 3.1. Growth Rate

We first determined the growth rates of the adult fibroblast culture C1 and the neonatal fibroblast culture C2 cultivated in medium containing glucose, galactose, or fructose as carbohydrate (Table 1). When cultivated in medium containing glucose, the doubling time of C1 fibroblasts was ~1.5 times longer than that of C2 fibroblasts. When cultivated in medium containing galactose, the doubling times of C1, as well as C2, increase ~3-fold compared to cultivation in medium containing glucose, whereas in medium containing fructose, the doubling times of C1 and C2 are only slightly longer than those in medium containing glucose.

### 3.2. Copy Number of mtDNA

To examine if the type of carbohydrate in the medium affects the cellular mtDNA level, we measured the mtDNA copy number per cell with ddPCR for the two fibroblast cultures cultivated in medium containing glucose, galactose, or fructose (Figure 1). When cultivated in medium containing galactose, mtDNA copy numbers of both cultures are, on average, slightly higher than when cultivated in medium containing glucose, but the differences do not reach statistical significance. However, when both cultures are cultivated in medium containing fructose, their mtDNA copy numbers show a further modest increase and are significantly higher than when cultivated in medium containing glucose (on average 19% higher for C1 and 25% higher for C2).

### 3.3. Levels of Nuclear and Mitochondrial Transcripts

To investigate whether the type of carbohydrate in the medium changes the transcript levels of genes involved in metabolism and mitochondrial biogenesis, we assessed the mRNA levels of a selection of representative genes by reverse transcriptase RT-qPCR. We started with the nuclear genes *NDUFS2* and *NDUFB7*, both coding for subunits of OXPHOS complex I, *COX4I1*, coding for a subunit of OXPHOS complex IV, *TOMM20*, coding for the translocase of the outer mitochondrial membrane subunit 20, and *GAPDH*, coding for the glycolytic enzyme glyceraldehyde-3-phosphate dehydrogenase (Figure 2a). Although transcript levels of these genes are, on average, always higher in cells cultivated in medium containing galactose or fructose than when cultivated in medium containing glucose, biological replicate experiments (*n* = 5–7) show large variations, rendering the differences in transcript levels not statistically significant. 

The two strands of mtDNA are transcribed into three polycistronic transcripts that are subsequently processed into separate rRNAs, tRNAs, and mRNAs [30]. Two overlapping polycistronic transcripts start at the initiation of transcription sites IT_H1_ and IT_H2_ to transcribe the H-strand, and one polycistronic transcript starts at the initiation of transcription site IT_L_ to transcribe the L-strand. Transcription of the H-strand starts most frequently at IT_H1_ and then terminates after transcription of the rRNA genes to yield the bulk of the rRNAs [38,39]. In contrast, transcription starting at IT_H2_ is less frequent but encompasses all H-strand genes. We assessed transcript levels of mitochondrial genes representing each of the three polycistronic transcripts: the 16S rRNA gene *MTRNR2* for the polycistronic transcripts starting at IT_H1_ or IT_H2_, the complex IV subunit gene *MTCO2* for the polycistronic transcript starting at IT_H2_, and the complex I subunit gene *MTND6* for the polycistronic transcript starting at IT_L_ (Figure 2b). Similar to the findings for the nuclear transcript levels, the mitochondrial transcript levels are, on average, always higher in cells cultivated in medium containing galactose or fructose than in cells cultivated in medium containing glucose (*n* = 5–6), but the differences are not statistically significant for the majority of transcripts. Only *MTCO2* transcript levels of C1 fibroblasts in medium containing galactose and *MTND6* transcript levels of C1 fibroblasts in medium containing galactose or fructose are significantly higher than when cultivated in medium containing glucose.

Finally, we investigated the transcript levels of the *PPARGC1A* gene encoding peroxisome proliferator-activated receptor γ co-activator (PGC-1α) and the *NRF1* gene encoding nuclear respiration factor 1 (NRF1; Figure 2c). PGC-1α governs mitochondrial biogenesis as a co-activator of several transcription factors, including NRF1, that orchestrate the coordinated transcription of key nuclear genes coding for mitochondrial proteins [40,41,42]. As observed in the earlier experiments, in most cases, *PPARGC1A* and *NRF1* transcript levels are, on average, higher in cultures cultivated in galactose- or fructose-containing medium than in cultures cultivated in glucose-containing medium but, due to the large variation of the results of biological repeats (*n* = 4–6), the differences are not statistically significant. 

**Figure 2 biology-12-01002-f002:**
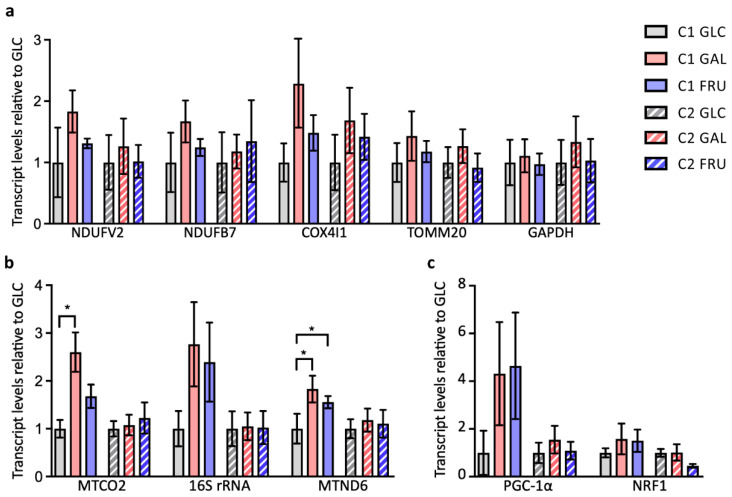
Transcript levels in the two cultures C1 and C2 grown with glucose (GLC), galactose (GAL), or fructose (FRU) determined with reverse transcriptase RT-qPCR. (**a**) Transcript levels of the nuclear structural genes *NDUFV2*, *NDUFB7*, *COX4I1*, *TOMM20,* and *GAPDH*. (**b**) Transcript levels of the mitochondrial structural genes *MTCO2*, *MTRNR2* (16S rRNA), and *MTND6*. (**c**) Transcript levels of the nuclear co-activator and transcription factor genes *PPARGC1A* (PGC-1α) and *NRF1*. The transcript levels in cells grown with galactose or fructose are shown as mean ± SEM relative to the transcript level in cells grown with glucose (*n* = 5–7). Asterisks indicate statistically significant differences (* *p* < 0.05; two-way ANOVA with two-tailed ratio paired Student’s *t*-test).

### 3.4. Protein Levels of Enzymes Involved in Glycolysis

Although the transcript analysis was largely inconclusive, we continued our systematic exploration of the effects of the three carbohydrates by looking at protein levels of key glycolytic enzymes on Western blots of SDS-denaturing gels (Figure 3a,c). Hexokinase 1 (HK1) is the most abundant hexokinase isoform that catalyses the first step of glycolysis [43]. Western blots reveal a large increase in HK1 levels in both C1 and C2 fibroblast cultures when cultivated in medium containing galactose or fructose compared to medium containing glucose, with a greater difference in C1 cells.

We also examined the protein levels of the glycolytic enzymes GAPDH and pyruvate kinase isoform M2 (PKM2; Figure 3a,c). GAPDH is regarded as a constitutively expressed housekeeping protein [44]. In line with this, GAPDH levels are unaffected by the type of carbohydrate in the medium. The PKM2 isoform is involved in the switch to aerobic glycolysis [45] and regulation of the metabolic flux in response to glucose restriction [46,47]. Notwithstanding this, PKM2 levels remained unchanged in the three different media.

Pyruvate, the end product of glycolysis, is converted into lactate by lactate dehydrogenase when it is not further processed in the mitochondria. Lactate is subsequently released from the cell through the monocarboxylate transporter 4 (MCT4) embedded in the plasma membrane. Although we do not detect substantial changes in MCT4 levels in C2 fibroblasts cultivated in the three different media and in C1 fibroblasts grown in the galactose-containing medium compared to glucose, MCT4 levels are lower in C1 fibroblasts when cultivated in fructose (Figure 3a,c).

**Figure 3 biology-12-01002-f003:**
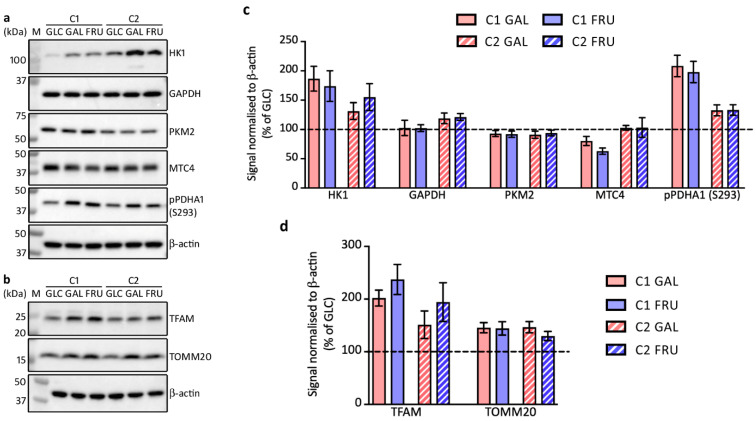
Western blot analysis of SDS-denaturing gels loaded with samples from the two cultures C1 and C2 grown with glucose (GLC), galactose (GAL), or fructose (FRU), probed with antibodies against glycolytic pathway (**a**) or mitochondrial marker proteins (**b**). Migration of protein markers (M) is indicated. (**c**) Mean ± SEM signals of glycolytic pathway proteins and (**d**) mitochondrial marker proteins normalised to β-actin and expressed as a percentage of the signal of cells grown with glucose (*n* = 3).

We also investigated the levels of pyruvate dehydrogenase E1 subunit a1 phosphorylated at serine residue 293 (pPDHA1 (S293)). PDHA1 is a subunit of pyruvate dehydrogenase, which is one of the three enzymatic components (E1–3) of the pyruvate dehydrogenase complex in the mitochondrial matrix. Pyruvate dehydrogenase catalyses the irreversible decarboxylation of pyruvate into acetyl-CoA. It provides the link between glycolysis in the cytosol and the Krebs cycle in the mitochondrial matrix. Pyruvate dehydrogenase activity is controlled by specific E1 kinase and phospho-E1-phosphatase enzymes. Phosphorylation of PDHA1 tyrosine residue 301 in tandem with phosphorylation of serine residue 293 inhibits pyruvate dehydrogenase activity [48,49]. Western blots indicate that pPDHA1 (S293) levels are higher in both fibroblast cultures when cultivated in medium containing galactose or fructose than when cultivated in medium containing glucose (Figure 3a,c). This suggests that pyruvate dehydrogenase activity is inhibited when cells are cultivated in galactose or fructose.

### 3.5. Protein Levels of Factors Involved in Mitochondrial Biogenesis

Mitochondrial transcription factor A (TFAM) is an essential activator of mitochondrial transcription and plays a crucial role in mtDNA maintenance [30]. Western blots show that TFAM levels are considerably higher in C1 fibroblasts when cultivated in medium containing galactose or fructose compared to cultivating in medium containing glucose (Figure 3b,d). On average, similar increases are also observed for C2 fibroblasts cultivated in medium containing galactose or fructose.

TOMM20 is an established marker of mitochondrial content (see, e.g., [50]). Western blots show modest increases in TOMM20 levels in C1 as well as C2 fibroblasts cultivated in medium containing galactose or fructose compared to cells cultivated in medium containing glucose (Figure 3b,d), suggesting a small increase in mitochondrial content.

### 3.6. Protein Levels of OXPHOS Complex Subunits

In the following experiments, we studied the levels of representative subunits of the OXPHOS complexes on Western blots of SDS-denaturing gels. First, we assessed the levels of several subunits encoded on the mtDNA: subunit MTCYB of complex III, subunits MTCO1 and MTCO2 of complex IV, and subunit MTATP6 of complex V (Figure 4a,c). We find increased steady-state levels of all the assessed subunits in C1 and C2 fibroblasts when cultivated with galactose or fructose compared to glucose, with the exception of MTCYB levels in C2 cells, which are not affected by the type of carbohydrate in the medium.

Next, we investigated the levels of a number of OXPHOS subunits encoded on the nuclear DNA: subunit NDUFB6 of complex I, subunit SDHA of complex II, subunit UQCRC2 of complex III, subunit COX4 of complex IV, and subunit ATPA5 of complex V (Figure 4b,d). Except for SDHA levels in C2 fibroblasts, the levels of these subunits follow a similar pattern to the mtDNA-encoded ones, presenting higher steady-state levels in both fibroblast cultures cultivated in medium containing galactose or fructose compared to cultivation in medium containing glucose.

**Figure 4 biology-12-01002-f004:**
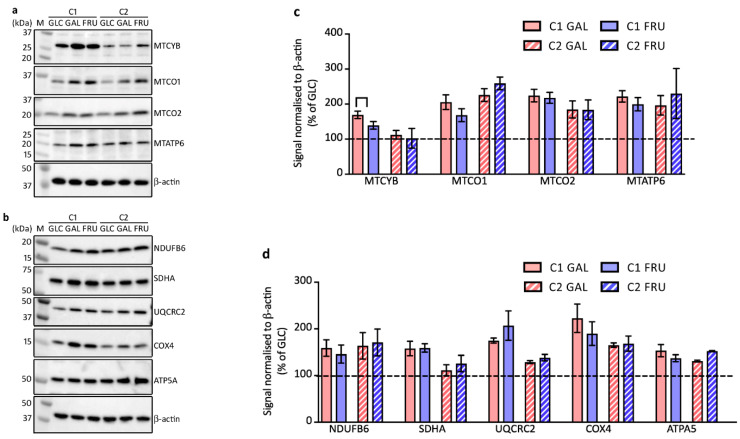
Western blot analysis of SDS-denaturing gels loaded with samples from the two cultures C1 and C2 grown with glucose (GLC), galactose (GAL), or fructose (FRU), probed with antibodies against mitochondrial- (**a**) or nuclear-encoded (**b**) OXPHOS subunits. Migration of protein markers (M) is indicated. Mean ± SEM signals of mitochondrial- (**c**) and nuclear-encoded (**d**) OXPHOS subunits normalised to β-actin and expressed as a percentage of the signal of cells grown in glucose (*n* = 3–5).

### 3.7. Levels of OXPHOS Complexes

After evaluation of the single subunits of the OXPHOS complexes on Western blots of SDS-denaturing gels, we determined the levels of all five holo-enzyme complexes on Western blots of blue-native gels (Figure 5a,b). The blots reveal that the levels of complex I, III, and IV, which together form the respirasome [51], are 2–5-fold higher in both fibroblast cultures cultivated in medium containing galactose or fructose compared to cultivation in high-glucose medium. Complex II and complex V levels are also higher but show a more modest increase in both fibroblast cultures. The blots probed with an antibody against complex V subunit ATP5A show free ATP5A in addition to holo-complex V (Figure 5a). Apparently, not all ATP5A is assembled in the complex.

### 3.8. Activities of Lactate Dehydrogenase, OXPHOS Complex IV, and Citrate Synthase

Lactate dehydrogenase activity assays show that the activity is ~20% higher in C1 fibroblasts cultivated in a galactose- or fructose-based medium than when cultivated in a glucose-based medium. On average, lactate dehydrogenase activity is also higher in C2 fibroblasts cultivated with galactose or fructose than when with glucose, but the differences do not reach statistical significance (Figure 6a). Spectrophotometric activity assays indicate that complex IV (cytochrome-*c* oxidase) activity is 50–80% higher in C1 and C2 fibroblasts cultivated in medium containing galactose or fructose compared to cultivation in medium containing glucose (Figure 6b). We also determined the activity of the Krebs cycle enzyme citrate synthase. The activity of this enzyme is often used as a biomarker of mitochondrial content in the cell [52,53,54]. The measurements show that citrate synthase activity is 12–20% higher in both fibroblast cultures cultivated in medium containing galactose or fructose than when cultivated in medium containing glucose (Figure 6c), suggesting a small increase in mitochondrial content.

### 3.9. Rates of Mitochondrial Respiration

In the final experiments, we measured mitochondrial respiration during mitostress tests on a Seahorse XFp platform (Figure 7a–c). The data indicate, on average, a ~60% increase in basal respiration in the two fibroblast cultures cultivated in medium containing galactose or fructose instead of glucose; however, this increase only reaches significance for C2 fibroblasts cultivated with fructose-based medium. In addition, 50–130% increases in spare respiratory capacity are observed in the cultures cultivated with galactose- or fructose-based medium. These increases are significant, except for C2 fibroblasts cultivated with fructose-based medium. Furthermore, significant 70–250% increases in maximal respiration are found in both cultures cultivated with galactose- or fructose-based medium, whereas the respiration linked to ATP production shows non-significant 35–55% increases. Substantial differences between cultivation in galactose- or fructose-containing medium are not observed. Thus, carbohydrate switching from glucose to either galactose or fructose induces a strong upsurge in maximal respiration.

## 4. Discussion

This study aimed to systematically characterise the remodelling of the mitochondrial bioenergetic pathways in fibroblasts cultivated in medium in which galactose or fructose was substituted for glucose. We find a strong increase in the activity of the OXPHOS complexes when fibroblasts are cultivated in galactose- or fructose-based mediums. Spare respiratory capacity, maximal respiration, and complex IV activity are all significantly higher. The enhanced activity appears to be the result of an increase in OXPHOS complexes levels because Western blots of blue-native gels demonstrate that cells exposed to galactose or fructose as their main carbohydrate source show markedly higher levels of fully assembled complex I, III, IV, and V. These findings are further supported by Western blots of SDS-denaturing gels, which indicates that galactose and fructose induce an increase in the levels of most individual subunits of the OXPHOS complexes. Our observations are in accordance with previous reports showing increased respiration and complex IV activity in muscle and cancer cell lines cultivated in galactose-based medium instead of glucose-based medium [15,22,24,55]. Higher levels of individual OXPHOS subunits have earlier been reported in muscle cells [25], cancer cell lines [22,24], and fibroblasts [22,23] cultivated in galactose-based medium.

We hypothesised that the changes in OXPHOS protein levels were regulated at the transcriptional level. Therefore, we measured the transcript levels of several OXPHOS subunit genes. Although transcript levels are, on average, higher in fibroblasts cultivated in galactose- or fructose-based media than when cultivated in glucose-based medium, we observe high variability of the values in biological replicate experiments. We think that, despite our standardised culture conditions, minor differences in nutrient availability and cell density have a major impact on the quickly adapting transcript levels, especially in proliferating cultures. We also examined the transcript levels of the master regulator of mitochondrial biogenesis PGC-1α and transcription factor NRF1 but the results were inconclusive due to the high variability of replicate experiments. We did observe, however, considerably higher TFAM protein levels in fibroblasts cultivated in galactose- or fructose-based medium instead of glucose-based medium. The promoter of the *TFAM* gene contains an NRF-1 responsive element, and the TFAM protein plays a key role in mtDNA transcription and stability [30,41].

In addition to the large increases in activity and OXPHOS complexes levels, our results suggest a small increase in mitochondrial content in fibroblasts cultivated with galactose- or fructose-based medium instead of glucose-based medium. Fructose induces a modest increase in mtDNA copy number, whilst both galactose and fructose lead to higher TOMM20 levels and citrate synthase activity. These results are corroborated by previous reports of higher mitochondrial content in muscle and cancer cell lines cultivated in galactose-based medium and appear to be accompanied by changes in mitochondrial morphology [22,24,26]. The modest increase in mitochondrial content is not reflected in a statistically significant increase in respiration coupled to ATP production. The reason for this might be that, unlike ddPCR or spectrophotometric enzyme assays, the respiration assays are not sensitive enough to detect small differences.

Although we did not conduct a comprehensive analysis of the effects of a carbohydrate switch in the medium on glycolysis, our Western blot analysis of some key enzymes of this pathway indicates an increase in HK1 levels in C1 and C2 fibroblasts cultivated in galactose- or fructose-based medium, a small decrease in MTC4 levels in C1 fibroblasts cultivated in fructose-based medium, and a small increase in lactate dehydrogenase activity in C1 fibroblasts cultivated in galactose-or fructose-based medium compared to glucose-based medium. GAPDH and PKM2 levels are unaffected. Others have documented that the levels of HK1, HK2, and the glucose transporter GLUT1 are unaffected by the substitution of galactose for glucose in the medium in HeLa cells [22]. While most glycolytic enzyme levels may not be directly affected, we find an increase in pPDHA1(S293) levels in fibroblasts cultivated in medium containing galactose or fructose, compared to cultivation in glucose-containing medium. Others report that PDHA1 levels do not change in fibroblasts cultivated in galactose-based medium instead of glucose-based medium [22]. The phosphorylation status of PDHA1 controls pyruvate dehydrogenase activity, which determines whether pyruvate generated by glycolysis is used in the Krebs cycle to provide NADH and FADH_2_ substrates for OXPHOS or is converted into lactate. Phosphorylation of PDHA1 at serine residue 293 inactivates pyruvate dehydrogenase [48,49]. Therefore, the higher pPDHA1(S293) levels suggest that pyruvate is not used to maintain the Krebs cycle. This fits with the notion that in galactose- or fructose-based mediums, glutamine is the main energy source, supplying anaplerotic replenishment of the Krebs cycle, whilst galactose and fructose are not consumed as fuel but are used to provide building blocks for nucleic acid synthesis [18,20].

All experiments were performed with two human primary dermal fibroblast cultures, a fast-growing culture (C2), derived from a neonatal donor, and a slower-growing culture (C1), derived from an adult donor. The cellular and metabolic responses to the use of galactose or fructose show only marginal differences between the two cultures, suggesting that both carbohydrates can be used to stimulate mitochondrial activity in in vitro models. Interestingly, while doubling times for both cultures triple when cultivated in galactose-based medium, they are only marginally longer when cultivated in fructose-based medium compared to glucose-based medium. A marked drop in cellular proliferation in galactose-based medium compared to glucose-based medium has also been observed by others for muscle and cancer cell lines [22,27,55].

## 5. Conclusions

Galactose- and fructose-based media strongly induce aerobic metabolism in dermal fibroblast cultures and result in a modest increase in mitochondrial content. Both media circumvent the Crabtree effect seen during cultivation in glucose-based mediums. Therefore, galactose- as well as fructose-based media are thought to enhance a mitochondrial phenotype of cultured fibroblasts derived from patients with partial mitochondrial defects. The slower growth rate in galactose-based medium can become a problem when naturally slow-growing fibroblasts from patients are cultivated. As galactose-based medium and fructose-based medium induce OXPHOS to a comparable level, we recommend using fructose-based medium to accentuate partial OXPHOS defects in cultured patients’ fibroblasts.

## Figures and Tables

**Figure 1 biology-12-01002-f001:**
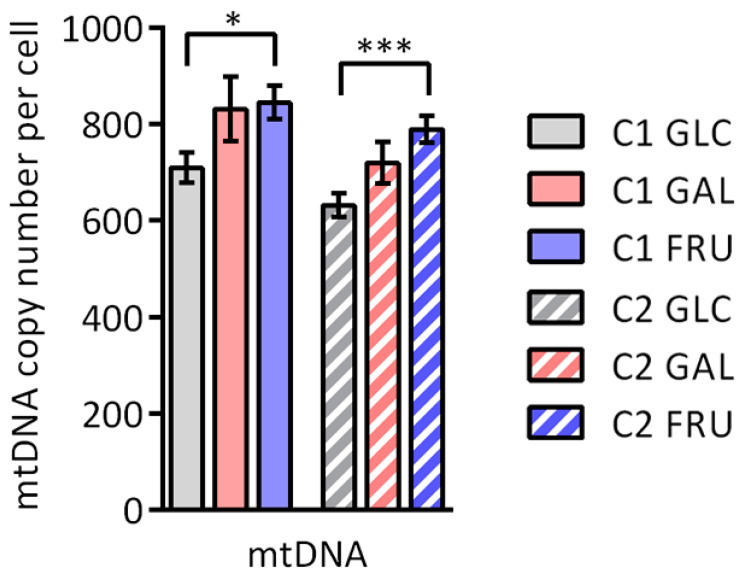
Copy number of mtDNA per cell of the two cultures C1 and C2 grown with glucose (GLC), galactose (GAL), or fructose (FRU) determined by ddPCR. The mtDNA copy number is shown as mean ± SEM (*n* = 8). Asterisks indicate statistically significant differences (* *p* < 0.05, *** *p* < 0.001; two-way ANOVA with Tukey’s multiple comparison test).

**Figure 5 biology-12-01002-f005:**
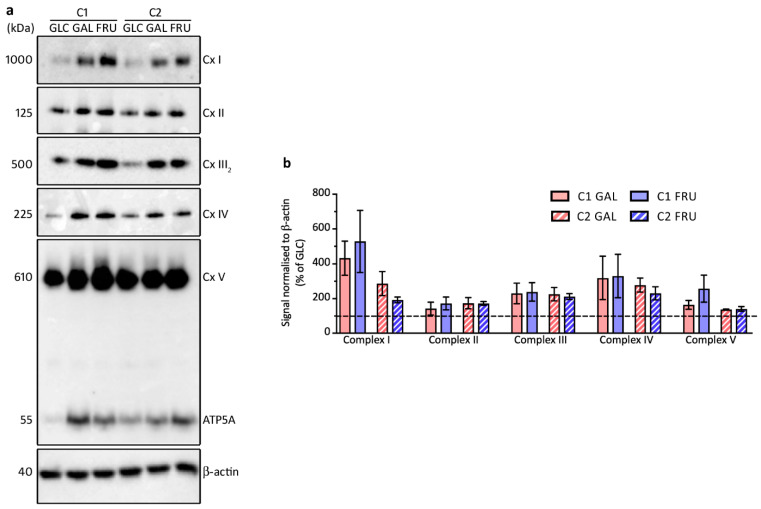
Western blot analysis of blue-native gels loaded with samples from the two cultures C1 and C2 grown with glucose (GLC), galactose (GAL), or fructose (FRU), probed for OXPHOS enzyme complexes. (**a**) Western blots probed with antibodies against subunits of each of the five enzyme complexes (Cx I–V). A western blot of an SDS-denaturing gel loaded with the same samples was probed for β-actin to confirm even loading. Molecular weights are indicated. (**b**) Mean ± SEM signals of the OXPHOS complexes normalised to β-actin and expressed as a percentage of the signal of cells grown with glucose (*n* = 3).

**Figure 6 biology-12-01002-f006:**
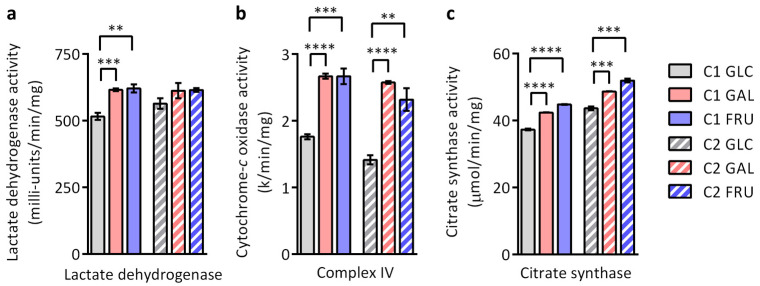
Enzyme activities of the two cultures C1 and C2 grown with glucose (GLC), galactose (GAL), or fructose (FRU) determined by spectrophotometry. (**a**) Lactate dehydrogenase activity (*n* = 4), (**b**) OXPHOS complex IV activity (*n* = 4), and (**c**) citrate synthase activity (*n* = 3) are shown as mean ± SEM. Asterisks indicate statistically significant differences (** *p* < 0.01, *** *p* < 0.001, **** *p* < 0.0001; two-way ANOVA with Tukey’s multiple comparison test).

**Figure 7 biology-12-01002-f007:**
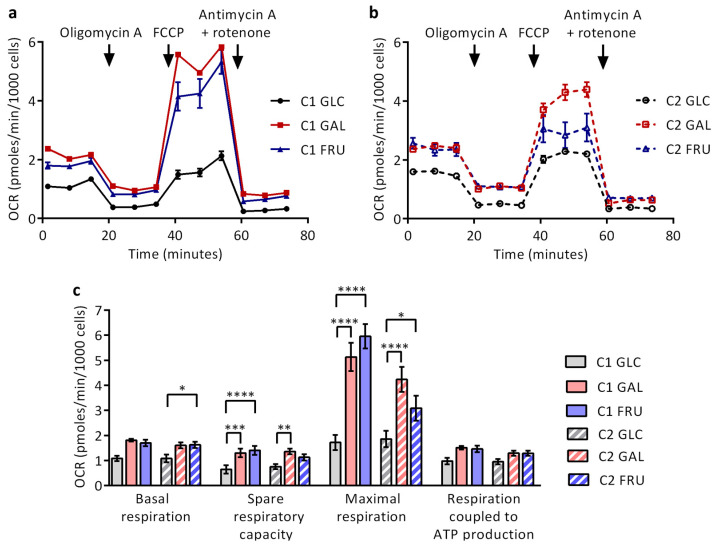
Mitochondrial respiration of the two cultures C1 and C2 grown with glucose (GLC), galactose (GAL), or fructose (FRU) examined on a Seahorse XFp platform. Oxygen consumption rate (OCR) profiles in mitostress tests of (**a**) fibroblasts C1 and (**b**) fibroblasts C2. Oligomycin A, FCCP, and rotenone + antimycin A were sequentially added at specific time points (arrows) to dissect mitochondrial respiratory function. Error bars denote standard deviation of technical triplicates. (**c**) Mean basal respiration, spare respiratory capacity, maximal respiration, and respiration coupled to ATP production normalised to cell number (*n* = 5–6). Error bars denote SEM. Asterisks indicate statistically significant differences (* *p* < 0.05, ** *p* < 0.01, *** *p* < 0.001, **** *p* < 0.0001; two-way ANOVA with Tukey’s multiple comparison test).

**Table 1 biology-12-01002-t001:** Doubling times ^1^ of cells grown with different carbohydrates.

Cell Culture	Glucose	Galactose	Fructose
C1 (adult fibroblasts)	37.2 ± 1.3 h	122.0 ± 6.0 h	40.8 ± 1.9 h
C2 (neonatal fibroblasts)	25.3 ± 1.1 h	66.2 ± 3.2 h	31.6 ± 1.4 h

^1^ Doubling times ± SEM (*n* = 3).

## Data Availability

The data presented in this study are available on request from the corresponding author.

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
