# Peer review of "Remodelling of the Mitochondrial Bioenergetic Pathways in Human Cultured Fibroblasts with Carbohydrates"

_biology, 2023, doi:10.3390/biology12071002_

Round 1
Reviewer 1 Report
The manuscript by Protasoni et al investigated changes in oxidative and glycolytic metabolism in primary fibroblasts cultured in the presence of glucose, galactose or fructose. Data are clearly presented. The study appears interesting but suffers of some major flaws that limit its value.
The authors measured the levels of GLUT1, what about the other glucose transporters? GLUT2 could be involved in galactose transport, and GLUT5 is mainly involved in fructose transport across the membrane. I would suggest the authors to check these transporters.
Considering the differences in the levels of proteins involved in glycolysis, and the differences in the levels of MCT4 I would suggest the authors to quantify the levels of lactate in culture media.
To complete data, I would also suggest to quantify extracellular acidification rate in both cell lines, in the presence of glucose, galactose, fructose.
What about the levels of ATP produced in all conditions?
Reviewer 2 Report
The research article “Remodelling of the mitochondrial bioenergetic pathways in human cultured fibroblasts with carbohydrates” by Margherita Protasoni reports that the use of galactose- or fructose-based medium can switch the fibroblasts’ energy metabolism from predominantly depending on glycolysis to a state more dependent on oxidative phosphorylation (OXPHOS). The authors cultured “young” (neonatal) and “old” primary fibroblast cultures under different conditions, in media supplemented with glucose, galactose or fructose. The authors observed that galactose and fructose-supplemented media stimulated switch to OXPHOS in both types of fibroblasts, while galactose also significantly decreased proliferation of the cells. Thus, culture in fructose based medium will switch fibroblasts, regardless of their “age”, to more OXPHOS-dependent state. This observation might be extrapolated with some degree of caution to other types of cells with similar metabolic profiles and will be instrumental when studying partial oxidative phosphorylation defects and beyond (for example, if experimental design requires such switch of the cellular energy production).
In my opinion the manuscript is flawless, it is clear, all the references cover most recent findings in the field.
The experimental design is well-thought, statistic analysis is appropriate. The weaknesses of the study and questions are listed below -
The authors used two samples of fibroblasts only (adult fibroblasts from only one donor), thus the observed behaviour of the cells might be based on the genetic features of particular donor rather than biology on fibroblasts per se.
I also tend to agree with authors that the transcript analysis was largely inconclusive.
How do the authors explain slight difference between neonatal and adult fibroblasts in terms of the change of the doubling time comparing glucose and galactose-supplemented media?
Round 2
Reviewer 1 Report
I understood that authors focused mainly on remodelling of the mitochondrial bioenergetic pathways, not the glycolytic pathway in the cytosol. However, they assessed some glycolytic enzymes (HK1, GAPDH, PKM2 and MTC4). For this reason, I asked for additional data, but authors did not address any comments raised by the reviewer.
Author Response
As we mentioned earlier, we focused mainly on remodelling of the mitochondrial bioenergetic pathways, not the glycolytic pathway in the cytosol, in our manuscript for Special Issue on Mitochondrial Metabolism and Function in Health and Disease. The reason for this is that, while switching from glucose to galactose is used in the literature to accentuate a mitochondrial defect in cultured fibroblast derived from patients with a mitochondrial disorder, the actual effects of carbohydrate switching on mitochondrial function in fibroblasts are not well documented. We agree with the reviewer that it would be interesting to study glycolysis in more detail, but this is less relevant to in vitro model studies of cultured fibroblast from patients with a mitochondrial disorder. We are sorry, if we have confused the reviewer by addition of the analysis of some glycolytic enzymes (HK1, GAPDH, PKM2 and MTC4) as a prelude to a more detailed study of glycolysis for publication in the future.